# Certainty in Uncertain Times: Dental Education during the COVID-19 Pandemic–A Qualitative Study

**DOI:** 10.3390/ijerph20043090

**Published:** 2023-02-10

**Authors:** Katja Goetz, Hans-Jürgen Wenz, Katrin Hertrampf

**Affiliations:** 1Institute of Family Medicine, University Hospital of Schleswig-Holstein, Campus Lübeck, 23562 Luebeck, Germany; 2Department of Prosthodontics, Propaedeutics and Dental Materials, University Hospital of Schleswig-Holstein, Campus Kiel Germany, 24105 Kiel, Germany; 3Department of Oral and Maxillofacial Surgery, University Hospital of Schleswig-Holstein, Campus Kiel Germany, 24105 Kiel, Germany

**Keywords:** COVID-19, dental education, digitalization, practical course, qualitative study, students’ experiences, teachers’ experiences

## Abstract

Background: The restrictions concerning social contact due to the COVID-19 pandemic implied a rethinking of teaching methods at universities in general, and for practice-oriented teaching such as dental education in particular. This qualitative study aimed to assess aspects of feelings of certainty and uncertainty during this specific education process, incorporating the perspectives of teaching staff and dental students. Methods: Qualitative methods based on interviews were used for data collection. Dental students from different academic years (second, third, fourth, and fifth) and teaching staff responsible for the content and implementation of courses within the dental curriculum were recruited. The data analysis was performed by qualitative content analysis. Results: A total of 39 dental students and 19 teaching staff participated. When students and staff dealt positively with this specific situation, certainty was achieved. The availability of presentations and clear communication enhanced feelings of certainty. The participants often felt unsure about how to handle such a challenging situation and felt insecure when planning for the semester. The students missed contact with other students and argued that the information policy on their dental studies was not transparent enough. In addition, dental students and teaching staff were nervous about the risk of infection from COVID-19, especially in practical courses with patient contact. Conclusions: The COVID-19 pandemic situation leads to a rethinking of dental education. Feelings of certainty can be strengthened by clear and transparent communication as well as training in online teaching methods. To reduce uncertainty, it is crucial to establish channels for information exchange and feedback.

## 1. Introduction

In March 2020, COVID-19 was declared by the World Health Organization as a pandemic disease, and this led to severe restrictions on social contact [1]. The social distancing requirements in everyday life also implied a rethinking of teaching methods at universities in general, and for practice-oriented teaching in particular.

Dental education is a very practice-oriented course of study. However, universities worldwide decided to offer a digital summer semester in 2020 to maintain the continuity of dental education [2,3]. As a result, the use of e-learning has significantly expanded. A survey between the end of March and the beginning of April 2020 by the Association of Dental Education in Europe showed that 90% of dental schools used online pedagogical software tools, 72% used live or streamed videos, and 48% provided links to further online materials [4]. However, dental education cannot only be taught digitally. It is a very practical study program. An essential element of the dental curriculum is early and continuous patient contact as part of clinical treatment [5,6,7]. Students learn competencies that are important for routine treatment processes [8].

Due to the relatively low incidence rates of COVID-19 in Germany’s northernmost federal state, senior staff at the Kiel Dental Clinic developed comprehensive social distancing and hygiene measures for the practical courses, and liaised with the relevant local health authority, university, and university hospital. This dental clinic was thus the first clinic in Germany to receive approval under special provision to conduct in-person practical courses with patients from the beginning of May 2020. Theoretical courses were also performed digitally.

Surveys in different countries show that students and clinical staff were relieved that dental education took place digitally. The suspension of face-to-face teaching, especially the practical courses with patients, severely contributed to the loss of students’ clinical competence [8,9,10]. Moreover, different studies show that the new teaching situation negatively impacts the mental health of dental students [11,12,13]. It is important to know more about how students and teaching staff receive certainty in times of uncertainty for the conceptualization of sustainable training concepts. 

Therefore, this study aimed to assess aspects of feelings of certainty and uncertainty during dental education, specifically during the COVID-19 pandemic, at the dental school in Kiel, Germany, by incorporating the perspectives of dental students and their teaching staff. 

## 2. Materials and Methods

### 2.1. Study Design

This study was designed as a qualitative interview study to assess the experiences of students and teaching staff regarding aspects of certainty and uncertainty, specifically during the COVID-19 pandemic. The consolidated criteria for reporting qualitative research (COREQ) were used [14].

### 2.2. Participants

Qualitative interviews were conducted with a purposive sample of students and teaching staff. Students from different academic years (second, third, fourth, and fifth) at the dental school in Kiel, Germany, were included in this study, along with associated teaching staff. Dental simulation courses took place in the fourth and sixth semesters, and clinical treatment courses took place from the seventh to tenth semesters. The eighth and tenth semesters were selected as examples for the treatment courses. The target sample consisted of 10 students from each of the four specialist semesters, along with the lecturers from the four departments and the departmental directors (n = 19 in total), taking into account theoretical saturation [15].

For students, the following inclusion criteria were applied: participants of the respective subject per semester, over 18 years of age, and sufficient knowledge of the German language. For lecturers, the inclusion criteria were: responsibility for teaching content and its implementation in one of the Dental Clinic’s four departments, over 18 years of age, and sufficient knowledge of the German language. If these criteria were not applied, the participants were excluded from the study. 

### 2.3. Setting

Students were recruited from courses held via video conference in the last third of the summer semester, and lecturers from the various departments were given personal presentations on the project. Participation was voluntary. Appointments for the interviews were then made in person or by email. Data collection took place between June and August 2020. All interviews were conducted by two female members of the working group (K.H.[dental practitioner background], K.G.[health services research background]), either in person or by telephone, and both were experienced in performing qualitative research. As described in the literature, no difference in data quality was observed between face-to-face and telephone interviews, and both may be recommended for use in the same qualitative study [16]. Third persons were not allowed in the interviews. The participants were grouped in such a way that they were unknown to their respective interviewers. All interviews and minuting adhered to the same predefined quality criteria, for example, with documentation of time, field notes, and of any issues or interruptions encountered during the interview. Socio-demographic data was requested from participants before the start of each interview.

### 2.4. Data Collection

An interdisciplinary team consisting of a sociologist, health services researcher, physician, and dental practitioners developed a semi-structured interview guide. After the final agreement of the guideline by the working group, the two interviewers went through the guideline step by step and agreed on the interview process. This coordination was repeated several times during the interviews. Following a literature review and discussion within the study team, the interview guide (as Appendix A) focused on two main topics:Feelings of certainty due to the experiences of teaching during the COVID-19 pandemic.Feelings of uncertainty due to the experiences of teaching during the COVID-19 pandemic.

An identical interview guide was used for both students and teaching staff which was tested with a student and a lecturer. The purpose for the tested interview guide was for comprehensibility and the sequencing of individual questions.

### 2.5. Data Analysis

All interviews were digitally audio recorded and transcribed in full verbatim. Transcripts were not submitted to participants for comments or correction. The texts were anonymized during transcription before undergoing qualitative content analysis [17]. For data analysis, the software ATLAS.ti 8.4 (Scientific Software Development GmbH, 2020) was used. For the generation of the thematic categories, the research team used a deductive-inductive approach. 

Firstly, a provisional category system was developed deductively based on the interview guidelines. Secondly, the provisional category system was then adjusted during analysis according to the content of the transcripts. Any new categories that emerged were then added following an inductive approach. Transcripts were coded independently into the main and subcategories by two female researchers, K.H. (dental practitioner background) and K.G. (health services research background), following intensive discussions that continued until consensus was achieved. Saturation was reached when, during the analyzing process, no new data were added [15]. Participant quotations were translated from German to English for publication purposes. The authors aimed to maintain reflexivity through K.H. and K.G. keeping notes on their thoughts, experiences, reflections, and feelings during the interviews, and discussed how their emotional reactions to participants influenced their interpretation of the results. This study’s quality was consistently ensured through adherence to predetermined quality standards. All interviews were conducted by the same two individuals, using the same interview guidelines for all students and teaching staff, under the same conditions. In addition, the proceedings from each interview were documented according to a previously agreed protocol.

### 2.6. Ethical Approval

This project was approved by the Ethics Committee of the University of Kiel, Germany (D509/20), and was conducted in accordance with the Declaration of Helsinki. Informed consent was obtained via a signed consent form, which included permission to publish anonymized quotes.

## 3. Results

Overall, 58 interviews were performed—39 with dental students and 19 with teaching staff. Participant characteristics are shown in Table 1 below. The interview duration varied and was about 31 min on average for the dental student group (min. = 22 min, max. = 50 min) and about 31 min for the teaching staff (min. = 15 min, max. = 41 min). 

The following sections describe the two main topics: “Certainty” and “Uncertainty”. Quotations are used to illustrate the relevant aspects reported by the participants (students [S] and teaching staff [TS]).

### 3.1. Main Topic: Certainty 

This topic describes what aspects were helpful in creating feelings of certainty in students and teaching staff during the COVID-19 pandemic and the changeover of teaching conditions. For this topic, two main categories were created and divided into different subcategories, as shown in Figure 1 below. 

The main category “Own experiences” observed aspects that contributed to the feeling of certainty. It emerged that a certain adjustment to this specific teaching situation was described by both students and teaching staff. Certainty was reached when acclimatization occurred in dealing with this specific situation, especially in applying the tools of online teaching. “I felt comfortable when I got through the first lectures and realized that I could cope well with them and work with them. That’s when I started to feel secure” (S36).

Students and teaching staff became experienced in their approach to the situation, as the following statement showed: “But after a time there was routine, so you could also assess whether you felt safe or not” (S14).

The second main category “Stabilizing aspects” comprised different elements that resulted in the feeling of certainty. Clear communication in such a specific situation was perceived as useful: “Everything was communicated clearly, so we were relatively sure how to design the course” (TS06). Students found the support of the teaching staff important for their own feelings of certainty: “Once the semester had been running for two or three weeks, and once you realized that you had the support of the teaching staff” (S04). The supervision of the teaching staff was also ensured during the practical courses under specific regulations, and contributed towards a stabilizing element of certainty: “I would say that I didn’t really feel insecure because care was guaranteed. I always had a lecturer who I could call on” (S35). Moreover, students stated that having regular online meetings, such as Zoom meetings with teaching staff, was a helpful aspect of feeling safe.

The availability of presentations was another aspect that enhanced certainty: “In terms of the events and the lectures… I wasn’t unsure. I thought to myself: Good, wonderful, I can manage it well” (S22). Students appreciated that the lectures were available on an online platform. Furthermore, the teaching staff also found this an important element for their own certainty: “As I said, I thought that through these online seminars, the students always had the opportunity to listen to the seminars again and again, so to speak, and the lectures again” (TS01). Some of the teaching staff were experienced in dealing with the technical aspects and the implementation of online teaching, and stated that this was important for their own feeling of certainty: “I actually felt relatively sure about the implementation of digitalization, the technical implementation, and the content implementation” (TS02).

### 3.2. Main Topic: Uncertainty

This topic describes which aspects led to students and teaching staff feeling uncertainty during the COVID-19 pandemic, as well as the change in teaching conditions. Three main categories were created and divided into different subcategories, as shown in Figure 2 below.

The main category “General aspects” comprised statements from the participants regarding how to handle this challenging situation that created general uncertainty during the COVID-19 pandemic, as the following statement from a member of the teaching staff illustrated: “The insecurity I had was due to the uncertainty of the situation. It wasn’t one with faulty or improvable behavior, so to speak, but the facts were simply not there and changed every day, and you had to adapt to the changing facts. And that caused the uncertainty” (TS13).

Students also felt uncertainty when dealing with such a specific situation: “Uncertainty was there at the beginning, where you didn’t know how it would continue, whether it would start, when it would start, how it would proceed” (S01). Furthermore, some students argued that the information policy about their dental studies was not transparent enough: “On the one hand, there was definitely a lack of communication. A lot of information, I think, that was also there, was not communicated to the students or somehow got to them via three corners” (S37). The general perception of the situation, especially the new teaching situation, was often characterized by the term ‘anxiety’ from students as well as teaching staff. The teaching staff who were responsible for the implementation of the specific regulations and hygienic conditions in the practical courses showed uncertainty concerning implementation and acceptance, as one statement demonstrated: “In the beginning, there was uncertainty in the sense that we were afraid that it wouldn’t work. We were afraid that we would fail with the measures we wanted to implement because not everyone would accept them” (TS03).

The main category “Missing aspects” included different issues that both students and teaching staff missed in their daily work. Both groups missed contact with students due to social distancing, and therefore a direct exchange of information or in-person lectures were not possible: “In such an event, where you interact with people in a diminished form, you somehow only have a very brief moment where everyone has the opportunity to ask or say something. I think that gives you a kind of security, when you have the feeling: OK, maybe someone has the same question” (S16).

Nearly all of the teaching staff stated that the change in teaching methods to online-teaching led to a lack of direct feedback during teaching: “What is uncertain and what I don’t have is feedback on the extent to which the lecture was received, the extent to which this mediation worked, and how the whole lecture was perceived” (TS02). The lack of knowledge as to whether the content of the lectures reached the students or not was perceived as a feeling of uncertainty by teaching staff: “With these purely digital lectures that you give, I don’t feel safe. It’s rather … Well, I don’t know if what I’m saying will be received” (TS11).

Some students stated that they missed a direct contact person during the online-teaching: “We felt insecure simply because we didn’t have a contact person in that sense” (S26). Due to the situation being unknown, students and teaching staff missed clear planning for the semester. This led to feeling uncertainty.

The teaching staff planned the first lockdown semester with a high level of insecurity, which was accompanied with great uncertainty: “All the planning we did was based on pure thought. We had to adapt to completely new schedules, we had to construct them. And there was a lot of uncertainty as to whether what we were planning for the whole semester was a plan that could be carried out precisely up to the last day. And if it didn’t fit, we would have had big problems” (TS13).

The main category of “Concerns” existed during the first lockdown semester, and included the risk of infection, especially in the practical courses, performance during the semester, design of exams, and implementation of online teaching. Students and teaching staff felt uncertain about the risk of being infected with COVID-19, especially in the practical courses with patient contact. “I think with this Corona issue, no one could block it out; certainly no one felt that way. A certain amount of uncertainty always remains, of course, and you also have your patient contacts” (TS11).

Moreover, one member of the teaching staff stated that the patients should to be tested for COVID-19 to provide more certainty: “In the beginning, I think we should have simply tested the patients. That would have given more security” (TS01). Students felt ambivalent about completing the practical course where patients are an integral component, and potentially infectious. “On the one hand, I was thinking: How can I in good conscience call elderly people to come to the clinic? At the time, one simply didn’t know how events would develop. And I tell my grandparents: Stay at home. And say to the elderly here: Come to the clinic. That was a bit of an ethical matter, but it was justifiable because we knew that the hygiene concept was right” (S21).

Only some students were concerned as to whether the semester would be completed or not. “I think the biggest uncertainty was whether the semester could take place at all. Digitalization was always just a bit of a side issue. I think the treatment course was central in all our minds” (S02). The majority of students and teaching staff were convinced that the semester would happen. Almost immediately, online tools were implemented for performing online teaching. However, this led to different feelings of uncertainty from the students and teaching staff. One member of the teaching staff stated: “Yes, it is different. The uncertainty was related to the way how I record the presentation or put it online practically, whether I’m not overwhelming the students with it or demanding more” (TS08).

Students were not sure about what kind of content could be provided within dental studies: “Yes, I think in the beginning there was uncertainty as to whether all the content could really be conveyed digitally, whether you would really get to grips with every topic in that sense” (S14). The technical implementation and use of different tools and devices were a challenge for some participants. “The first few times I logged into Zoom events, I sometimes didn’t know if it was the right thing to do, or if I had to click on it, or if I was going to miss it” (S11).

## 4. Discussion

The results show that different aspects could influence feelings of certainty and uncertainty during such a specific education process during the COVID-19 pandemic. The perspectives of dental students and their teaching staff are considered, and the various statements show that a process from uncertainty to certainty could be observed. As our results demonstrate, clear communication led to feelings of certainty and was also found in other studies [9]. However, the unclear information policy at the beginning of the online teaching situation and the practical courses under specific regulations was due to the situation itself. Never before have people experienced such a pandemic; they could not anticipate the consequences.

Different studies with dental students and their educators show the challenges during dental education and their effect on clinical performance [9,10,18]. In most countries, dental education changed to online education, which was perceived positively by dental students [4,18]. On the one hand, social distancing in terms of online teaching could minimize the infection rate of COVID-19 and lead to a feeling of certainty. On the other hand, social distancing could lead to the loss of a supportive network and create stress for students. Social support, such as by teaching staff, was found to be an important stabilizing element and was also observed as a protective factor concerning emotional loneliness [19,20].

Face-to-face exchanges and communication allow people to read non-verbal signals, but with social distancing, these non-verbal signals are absent, which could lead to a difficult communication process. The direct feedback that helps to assess whether a peer understands the meaning of the statement could lead to a sense of uncertainty in the communication process. Furthermore, it was found that the lack of peer feedback could have a negative impact on the effectiveness of online learning [21]. It can be assumed that the uncertainties over the outcome of the pandemic could have an effect on teaching staff as well as students’ well-being.

As already mentioned, the practical courses with patients started at the beginning of May 2020 with the concept of rigorous hygiene. However, this was associated with two main concerns: fear of infection, and completing the practical courses during the semester. Both concerns led to feelings of uncertainty for students and the teaching staff. Different studies show that the risk of COVID-19 infection was one of the reasons to perform any kind of education digitally [9,10,22,23]. From our perspective, we have not seen any students or teaching staff infected by COVID-19 within the practical courses during the first COVID-19 semester. Uncertainty has been replaced by a sense of security. Moreover, sustainable training concepts such as the field of technical competencies and the strengthening of mental health are necessary to become resilient in uncertain times.

### Limitations

This study has several limitations. The results cannot be generalized without further research because of the design of the qualitative study. Self-reporting comments from the students and teaching staff were used to present the results of this qualitative study. It is therefore not possible to make any assessments about the accuracy of the information. Moreover, participation in the interviews was voluntary. It must also be assumed that the study attracted interested students and teaching staff who were more open to the topics discussed. This “positive selection bias” may be reflected in the results and thus needs to be taken into account during interpretation.

## 5. Conclusions

The COVID-19 pandemic situation is a challenge for dental education, especially for the practical courses. Different aspects seem important for a sense of certainty and should be considered for future teaching situations. Clear and transparent communication would be useful, as well as training in online teaching methods, to strengthen the feeling of certainty. To reduce uncertainty in a pandemic situation, it is particularly important to establish channels for communication, information exchange, and feedback. In addition, this study provides empirical evidence and insight into the aspects that lead to feelings of certainty and uncertainty by dental students and their teaching staff during the COVID-19 crisis.

## Figures and Tables

**Figure 1 ijerph-20-03090-f001:**
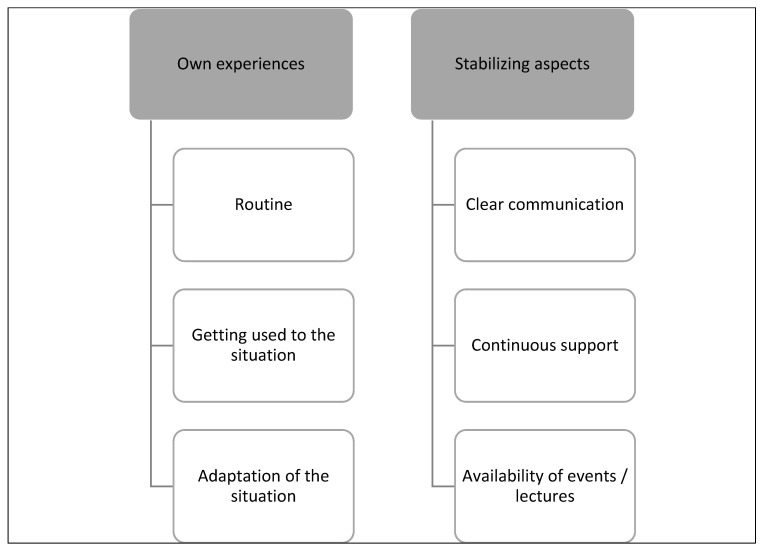
Certainty—main categories and subcategories.

**Figure 2 ijerph-20-03090-f002:**
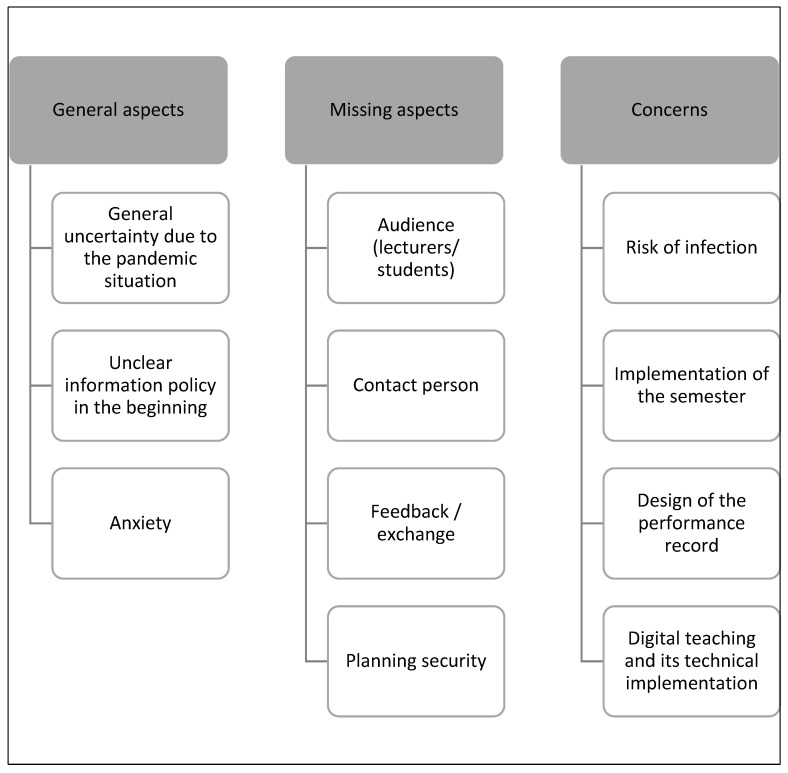
Uncertainty—main categories and subcategories.

**Table 1 ijerph-20-03090-t001:** Distribution of participants (n = 58).

Variable	Students (n = 39)	Lecturers (n = 19)
Women	27	6
Men	12	13
Age (Mean) (Range of ages)	25.2 (20–31)	44.0 (31–65)
Apprenticeship	13	-
Further completed study	5	-
Additional qualifications	5	-
Director of the clinic	-	4
Course instructor	-	7
Course assistant	-	8
Use of hardware *:		
Laptop	30	18
Tablet	10	0
Mobile phone Stationary PC	3 2	0 2
Access to camera (yes)	39	17
Access to microphone (yes)	38	17
Permanent availability	38	17
Adequate internet connection (yes)	38	19
Course in home office		2

* Multiple answers possible.

## Data Availability

The datasets generated during the current study are not publicly available due to the ethical requirements, but are available from the corresponding author upon reasonable request.

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
