# Peer review of "Certainty in Uncertain Times: Dental Education during the COVID-19 Pandemic–A Qualitative Study"

_ijerph, 2023, doi:10.3390/ijerph20043090_

Round 1

Reviewer 1 Report

- Line 44 requires English language check. 

- Please be more specific regarding the year the students were, not the semester. 

- Please explain the target sample of your study, have you performed any power analysis? 

- Please discuss your exclusion criteria

- Were the two interviewers (KH, KG) calibrated before the beginning of the data collection? 

- This is not a "dental education" related study but refers more to Psychology and the psychological situation of dental students during the pandemic. It would be very interesting if it was focused more to online versus in person dental education and the pandemic situation.

- The Results section needs to be placed in proper order so the readers would be able to understand it. 

Author Response

Reviewer 1

Comments and Suggestions for Authors

- Line 44 requires English language check. 

[Response]

- Please be more specific regarding the year the students were, not the semester. 

[Response]

We used the year of the students and changed it within the manuscript.

- Please explain the target sample of your study, have you performed any power analysis? 

[Response]

We performed no power analysis. The current study was a qualitative study and we used a purposive sample strategy. The target sample of our study is described within the section Materials and methods especially under the point participants.

- Please discuss your exclusion criteria

[Response]

We added one sentence within the sub-section Participants as follow:

“If these criteria were not applied the participants were excluded from the study.”

- Were the two interviewers (KH, KG) calibrated before the beginning of the data collection? 

[Response]

We did not assessed clinical variables. Therefore, a calibration was not required. We added within the subsection Data collection more information as follow.

“After the final agreement of the guideline by the working group, the two interviewer went through the guideline step by step and agreed on the interview process. This coordination was repeated several times during the interviews.”

- This is not a "dental education" related study but refers more to Psychology and the psychological situation of dental students during the pandemic. It would be very interesting if it was focused more to online versus in person dental education and the pandemic situation.

[Response]

Our manuscript addressed two topics from this special issue "COVID-19 in Dentistry and Dental Education". These are “Dental education and COVID-19” and “Problems faced by dental students during the pandemic”. During this specific education process such as expansion of e-learning and specific condition for practical courses experiences can be derived which lead to feeling of certainty and uncertainty. Therefore, we thought that our study is related to “dental education” and could be important for future teaching situation and strengthen the lectures as well students.

- The Results section needs to be placed in proper order so the readers would be able to understand it. 

[Response]

We had order the Results section on the basis of our main categories respectively their associated sub categories.

Reviewer 2 Report

Pandemic situation in 2020 led to significant expansion of e-learning. This situation caused multiple issues for students and academic staff in practice-oriented disciplines such as dentistry. The quantitative study provided by the authors aimed to assess aspects of feelings of 13 certainty and uncertainty during this specific education process, incorporating the perspectives of 14 teaching staff and dental students. Within the study 39 dental students and 19 teaching staff were interviewed. The authors provided the recommendations how to strength feeling of certainty and reduce uncertainty.  

In general, this survey study is interesting and falls into aim & scopes of the special issue.  

In the Introduction section the significance of the topic is clearly shown.

The study design meets the objective of the study. Also several details of the study design should be clarified to understand if possible bias are avoided:

1.       Participants. Did you calculate the sample size before the study (if applicable)? Is the sample of 39 students and 19 teachers enough to get significant results?

2.       Setting. Lines 88-89. Please mention how many invitations were send and how many respondents refused to participate in the survey. If applicable, mention the reason of refusal.

3.       Setting. Line 90. Please add more information related to interviewers such as credentials and occupation, relationship with participants.

4.       Setting. Line 91. In case of in person interview, was anyone else present besides the interviewer and participant?

5.       Data collection. Please provide the questions included into the interview guide. Did you made field notes?

In the Results section the major themes are clearly presented, the main findings are consistent with the data and supported by the quotations. Nevertheless minor themes are not addressed. If possible, describe diverse cases.

In the Discussion section it would be useful to give any ideas related to further investigations in the topic of e-learning in dentistry.

Author Response

Reviewer 2

Comments and Suggestions for Authors

Pandemic situation in 2020 led to significant expansion of e-learning. This situation caused multiple issues for students and academic staff in practice-oriented disciplines such as dentistry. The quantitative study provided by the authors aimed to assess aspects of feelings of 13 certainty and uncertainty during this specific education process, incorporating the perspectives of 14 teaching staff and dental students. Within the study 39 dental students and 19 teaching staff were interviewed. The authors provided the recommendations how to strength feeling of certainty and reduce uncertainty. 

In general, this survey study is interesting and falls into aim & scopes of the special issue. 

In the Introduction section the significance of the topic is clearly shown.

The study design meets the objective of the study. Also several details of the study design should be clarified to understand if possible bias are avoided:

  1. Participants. Did you calculate the sample size before the study (if applicable)? Is the sample of 39 students and 19 teachers enough to get significant results?

[Response]

This was a qualitative study which not aimed to get significant results but rather to know more about the subjective perspective on this specific situation. We did not calculate the sample size. We used a purposive sample strategy under consideration of theoretical saturation. We described this process within the section Materials and methods.

  1. Setting. Lines 88-89. Please mention how many invitations were send and how many respondents refused to participate in the survey. If applicable, mention the reason of refusal.

[Response]

The participants were recruited and invited to participate due to personal address. Nobody refused his or her participation.

  1. Setting. Line 90. Please add more information related to interviewers such as credentials and occupation, relationship with participants.

[Response]

More information to the related interviewers were added. Moreover we added within the subsection Setting:

“The participants were so grouped that they were unknown for the respective interviewers.”

  1. Setting. Line 91. In case of in person interview, was anyone else present besides the interviewer and participant?

[Response]

We added within the subsection following sentence:

“Third persons were not allowed in the interviews.”

  1. Data collection. Please provide the questions included into the interview guide. Did you made field notes?

[Response]

Please see supplementary file for the question included into the interview guide. Field notes were made. We added this aspect within the subsection Setting.

In the Results section the major themes are clearly presented, the main findings are consistent with the data and supported by the quotations. Nevertheless minor themes are not addressed. If possible, describe diverse cases.

[Response]

We found no diverse cases. Therefore, we cannot address this aspect.

In the Discussion section it would be useful to give any ideas related to further investigations in the topic of e-learning in dentistry.

[Response]

Thank you for this advice. Ideas were given under the Conclusion section.